# A Single Sacral-Mounted Inertial Measurement Unit to Estimate Peak Vertical Ground Reaction Force, Contact Time, and Flight Time in Running

**DOI:** 10.3390/s22030784

**Published:** 2022-01-20

**Authors:** Aurélien Patoz, Thibault Lussiana, Bastiaan Breine, Cyrille Gindre, Davide Malatesta

**Affiliations:** 1Institute of Sport Sciences, University of Lausanne, 1015 Lausanne, Switzerland; davide.malatesta@unil.ch; 2Volodalen Swiss Sport Lab, Research and Development Department, 1860 Aigle, Switzerland; thibault@volodalen.com (T.L.); bastiaan@volodalen.com (B.B.); cyrille@volodalen.com (C.G.); 3Volodalen, Research and Development Department, 39270 Chavéria, France; 4Research Unit EA3920 Prognostic Markers and Regulatory Factors of Cardiovascular Diseases and Exercise Performance, Health, Innovation Platform, University of Franche-Comté, 25000 Besançon, France; 5Department of Movement and Sports Sciences, Ghent University, 9000 Ghent, Belgium

**Keywords:** gait analysis, biomechanics, sensor, accelerometer

## Abstract

Peak vertical ground reaction force ($F_{z,\max}$), contact time ($t_{c}$), and flight time ($t_{f}$) are key variables of running biomechanics. The gold standard method (GSM) to measure these variables is a force plate. However, a force plate is not always at hand and not very portable overground. In such situation, the vertical acceleration signal recorded by an inertial measurement unit (IMU) might be used to estimate $F_{z,\max}$, $t_{c}$, and $t_{f}$. Hence, the first purpose of this study was to propose a method that used data recorded by a single sacral-mounted IMU (IMU method: IMUM) to estimate $F_{z,\max}$. The second aim of this study was to estimate $t_{c}$ and $t_{f}$ using the same IMU data. The vertical acceleration threshold of an already existing IMUM was modified to detect foot-strike and toe-off events instead of effective foot-strike and toe-off events. Thus, $t_{c}$ and $t_{f}$ estimations were obtained instead of effective contact and flight time estimations. One hundred runners ran at 9, 11, and 13 km/h. IMU data (208 Hz) and force data (200 Hz) were acquired by a sacral-mounted IMU and an instrumented treadmill, respectively. The errors obtained when comparing $F_{z,\max}$, $t_{c}$, and $t_{f}$ estimated using the IMUM to $F_{z,\max}$, $t_{c}$, and $t_{f}$ measured using the GSM were comparable to the errors obtained using previously published methods. In fact, a root mean square error (RMSE) of 0.15 BW (6%) was obtained for $F_{z,\max}$ while a RMSE of 20 ms was reported for both $t_{c}$ and $t_{f}$ (8% and 18%, respectively). Moreover, even though small systematic biases of 0.07 BW for $F_{z,\max}$ and 13 ms for $t_{c}$ and $t_{f}$ were reported, the RMSEs were smaller than the smallest real differences [$F_{z,\max}$: 0.28 BW (11%), $t_{c}$: 32.0 ms (13%), and $t_{f}$: 32.0 ms (30%)], indicating no clinically important difference between the GSM and IMUM. Therefore, these results support the use of the IMUM to estimate $F_{z,\max}$, $t_{c}$, and $t_{f}$ for level treadmill runs at low running speeds, especially because an IMU has the advantage to be low-cost and portable and therefore seems very practical for coaches and healthcare professionals.

## 1. Introduction

Running is defined as a cyclic alternance of support and flight phases, where at most one limb is in contact with the ground. Indeed, Novacheck [1] postulated that the presence of this flight phase ($t_{f}$) marks the distinction between walking and running gaits. In other words, the duty factor, i.e., the ratio of ground contact time ($t_{c}$) over stride duration, should be under 50% to observe a running gait [2,3]. Though running provides many health benefits, it is also associated with lower limb injuries [4,5], with a yearly incidence of running related injuries of up to 85% across novice to competitive runners [6,7]. Several biomechanical variables such as the peak vertical ground reaction force ($F_{z,\max}$, i.e., the maximum of the vertical ground reaction force during stance) and $t_{c}$ were used to identify runners at risk of developing a running related injury [8,9,10,11,12,13]. In fact, $F_{z,\max}$ is representative of the magnitude of external bone loading during the stance running phase while $t_{c}$ measures the time during which this force is applied [14]. Therefore, $F_{v,\max}$, $t_{c}$, and $t_{f}$ are key variables of running biomechanics.

A force plate is the gold standard method (GSM) to measure $F_{z,\max}$, $t_{c}$, and $t_{f}$. However, a force plate could not always be available and used [15,16]. In such a case, alternatives would be to use a motion capture system [17,18] or a light-based optical technology [19]. Nevertheless, even though these three systems can be used outside the laboratory [20,21,22], they suffer from a lack of portability and are restricted to a specific and small capture volume. To overcome such limitation, techniques were developed to estimate $F_{z,\max}$, $t_{c}$, and $t_{f}$ using portable tools such as inertial measurement units (IMUs), which are low-cost and practical to use in a coaching environment [23].

$F_{z,\max}$ was previously estimated using the vertical acceleration signal recorded by a sacral-mounted IMU [24,25]. For instance, a root mean square error (RMSE) of 0.15 BW was reported when using a machine learning algorithm that used data filtered using a 10 Hz 8th order low-pass Butterworth filter [25]. Another method calculated the center of mass and sacral marker vertical accelerations from their corresponding three-dimensional (3D) kinematic trajectories, and reported an RMSE ≤ 0.17 BW when estimating $F_{z,\max}$ from these acceleration signals [26]. The whole-body center of mass acceleration calculated from the kinematic trajectories was also used by Pavei, et al. [27] to estimate $F_{z,\max}$, but for a single participant, and by Verheul, et al. [28] to estimate the resultant ground reaction force impact peak (within the first 30% of the stance). Pavei, Seminati, Storniolo and Peyré-Tartaruga [27] reported an RMSE of ~0.15 BW for running speeds ranging from 7 to 20 km/h, while an error of ~0.20 BW was reported by Verheul, Gregson, Lisboa, Vanrenterghem and Robinson [28] for speeds between 7 and 18 km/h.

$t_{c}$ and $t_{f}$ are calculated from foot-strike (FS) and toe-off (TO) events, which can themselves be identified using different available techniques that used IMU data [24,25,29,30,31,32,33,34,35,36,37,38,39]. When using a sacral-mounted IMU, which is a natural choice as it approximates the location of the center of mass [40], either the forward [31] or the vertical acceleration [24,25] were used to estimate $t_{c}$ and $t_{f}$. On the one hand, Lee, Mellifont and Burkett [31] detected specific spikes in their unfiltered forward acceleration signals sampled at 100 Hz to identify FS and TO events. On the other hand, the vertical ground reaction force was estimated from the vertical acceleration signal recorded by the IMU (using Newton’s second law), which allowed detecting FS and TO events using a 0 N threshold [24,25]. A 5 Hz low-pass Butterworth filter (8th order) was shown to result in the best correlation between $t_{c}$, obtained from GSM and IMU data (sampled at 500 Hz) [24], while a machine learning algorithm that used data filtered using a 10 Hz 8th order low-pass Butterworth filter, resulted in an RMSE of 11 ms for $t_{c}$ [25]. The vertical acceleration (sampled at 208 Hz) was also used to estimate the effective contact and flight times [39], two variables that allow deciphering the landing-take-off asymmetry of running [41,42,43]. The authors estimated these effective timings by using a body weight threshold instead of a 0 N threshold, which allowed detecting effective FS and TO events and thus estimating effective contact and flight times. Moreover, the vertical acceleration was filtered using a Fourier series truncated to 5 Hz instead of the usual low-pass Butterworth filter. The authors reported an RMSE ≤ 22 ms for both effective contact and flight times.

As previously stated, more research investigating the effect of different filtering methods are needed when estimating biomechanical variables such as $F_{z,\max}$ and $t_{c}$ [24], especially because the low-pass cutoff frequency could affect the estimation of biomechanical variables [44,45]. For this reason, the first purpose of this study was to estimate $F_{z,\max}$ using a Fourier series truncated to 5 Hz to filter the acceleration signal recorded by a sacral-mounted IMU (IMU method: IMUM). The second aim of this study was to estimate $t_{c}$ and $t_{f}$ using the same filtered acceleration signal. We previously used this filter to estimate both effective contact and flight times [39], but this filter has never been used, to the best of the authors knowledge, to estimate $F_{z,\max}$, $t_{c}$, and $t_{f}$. In the present study, $t_{c}$ and $t_{f}$ were estimated from FS and TO events, themselves detected by modifying the body weight threshold we previously used [39]. We hypothesized that (i) an RMSE smaller than or equal to the 0.15 BW reported in Alcantara, Day, Hahn and Grabowski [25] should be obtained for $F_{z,\max}$, even if the IMUM is a simple method which does not rely on machine learning, as was the 3D kinematic method [26], and (ii) $t_{c}$ and $t_{f}$ should have an RMSE smaller than or equal to that we previously reported for effective contact and flight times (i.e., 0.22 ms) [39].

## 2. Materials and Methods

### 2.1. Participant Characteristics

One hundred recreational runners, which consisted of 27 females (age: 29 ± 7 years, height: 169 ± 5 cm, body mass: 61 ± 6 kg, and weekly running distance: 22 ± 16 km) and 73 males (age: 30 ± 8 years, height: 180 ± 6 cm, body mass: 71 ± 7 kg, and weekly running distance: 38 ± 24 km), were randomly selected from an existing database consisting of 115 participants [26] for the purpose of the present study. Participants voluntarily participated in this study, and to be included, they were required to run at least once a week and to not have current or recent lower-extremity injury (≤1 month). The study protocol was conducted according to the guidelines of the latest declaration of Helsinki and approved by the local Ethics Committee of the Vaud canton (CER-VD 2020-00334). Written informed consent was obtained for all subjects involved in the study.

### 2.2. Experimental Procedure and Data Collection

The experimental procedure and data collection has already been described elsewhere [39]. Briefly, an IMU of 9.4 g (Movesense sensor, Suunto, Vantaa, Finland) was firmly attached to the sacrum of participants using an elastic strap belt (Movesense, Suunto, Vantaa, Finland; see first figure in [39]). Then, after a warm-up run of 7-min, performed between 9 and 13 km/h on an instrumented treadmill (Arsalis T150–FMT-MED, Louvain-la-Neuve, Belgium), three 1-min running trials using speeds of 9, 11, and 13 km/h were recorded in a randomized order. Three-dimensional IMU and kinetic data corresponding to the first 10 strides following the 30-s mark of the running trials were kept for data analysis. Kinetic and IMU data were not exactly synchronized. However, the synchronization delay between kinetic and IMU data was small (≤50 ms). Therefore, kinetic and IMU data corresponded to the same 10 strides.

IMU data (saturation range: ±8 g) were collected at 208 Hz (manufacturing specification) using a home-made iOS application running on an iPhone SE (Apple, Cupertino, CA, USA). The IMU orientation was such that its medio-lateral (pointing towards the right side of the IMU), posterior–anterior, and inferior–superior axes were denoted as the *x*-, *y*-, and *z*-axis, respectively. These IMU data were transferred to a personal computer for post processing.

The force plate embedded into the treadmill together with the Vicon Nexus software (v2.9.3, Vicon, Oxford, UK) were used to collect kinetic data (200 Hz). In the laboratory coordinate system (LCS), medio-lateral (pointing towards the right side of the body), posterior–anterior, and inferior–superior axes were denoted as the *x*-, *y*-, and *z*-axis, respectively. The Visual3D Professional software (v6.01.12, C-Motion Inc., Germantown, MD, USA) was used to process the 3D ground reaction forces (analog signal), which were first exported in .c3d format. Then, the forces were low-pass filtered at 20 Hz using a 4th order Butterworth filter.

### 2.3. Gold Standard Method

For each running trial, FS and TO events were identified within Visual3D. These events were detected by applying a 20 N threshold to the vertical ground reaction force [46]. $t_{c}$ and $t_{f}$ were defined by the time between FS and TO events and between TO and FS events, respectively, while $F_{z,\max}$ was defined by the maximum of the vertical ground reaction force between FS and TO events and was expressed in body weights.

### 2.4. Inertial Measurement Unit Method

The custom c++ code [47] used to process IMU data has already been described elsewhere [39]. Briefly, the average angle between the *z*-axis of the IMU and LCS was calculated using the median values of the 3D raw acceleration data filtered using a truncated Fourier series to 0.5 Hz in each dimension. This angle was used to align (reorient) the *z*-axis of the IMU with the LCS. This reorientation process was not considered in previous research that used a sacral-mounted IMU to estimate $F_{z,\max}$ and $t_{c}$ [24,25,31]. Then, 3D reoriented data were filtered using a truncated Fourier series to 5 Hz in each dimension. The vertical ground reaction force was approximated by the filtered vertical acceleration signal multiplied by body mass. FS and TO events were detected using a 20 N threshold, which allowed to estimate $t_{c}$ and $t_{f}$. This 20 N threshold replaced the body weight threshold used in the original custom code described in [39] and is the only change made herein. $F_{z,\max}$ was estimated as the maximum of the approximated vertical ground reaction force between FS and TO events.

### 2.5. Data Analysis

The RMSE, both in absolute (ms and BW) and relative units, i.e., normalized by the corresponding mean value over all participants and obtained using the GSM, was calculated for $F_{z,\max}$, $t_{c}$, and $t_{f}$ averaged over the 10 analyzed strides for each participant and each running trial. Data analysis was performed using Python (v3.7.4, available at http://www.python.org (accessed on 25 October 2021)).

### 2.6. Statistical Analysis

All data are presented as mean ± standard deviation. To examine the presence of systematic bias on $F_{z,\max}$, $t_{c}$, and $t_{f}$ obtained from the GSM and IMUM for each speed, Bland–Altman plots were constructed [48,49]. In case of a systematic bias, a positive value indicates an overestimation of the IMUM compared to the GSM, while a negative value indicates an underestimation. In addition, lower and upper limit of agreements and 95% confidence intervals were calculated. The limits of agreements were calculated as the bias ± the smallest real difference (SRD). SRD defines the smallest change that indicates a clinically important difference and is calculated as SRD=1.96 σ, where σ is the standard deviation of the difference between the gold standard and estimated values [50,51]. Besides, a significant slope of the regression line indicates the presence of a proportional bias (heteroscedasticity). Then, as no obvious deviations from homoscedasticity and normality were observed in the residual plots, two-way [method of calculation (GSM vs. IMUM) × running speed (9 vs. 11 vs. 13)] repeated measures ANOVA using Mauchly’s correction for sphericity were performed for $F_{z,\max}$, $t_{c}$, and $t_{f}$. Holm corrections were employed for pairwise post hoc comparisons. The differences between the GSM and IMUM were quantified using Cohen’s *d* effect size, where |*d*| values close to 0.01, 0.2, 0.5, and 0.8 reflect a very small, small, moderate, and large effect size, respectively [52]. Statistical analysis was performed using Jamovi (v1.2, https://www.jamovi.org (accessed on 25 October 2021)) with a level of significance set at *p* ≤ 0.05.

## 3. Results

The raw forward acceleration recorded by the IMU and the filtered vertical acceleration recorded by the IMU, as well as the vertical acceleration recorded by the force plate during a running stride for three representative participants running at 11 km/h, are depicted in Figure 1.

Systematic biases (average over running speeds) were obtained for $F_{z,\max}$ (0.07 BW) and $t_{c}$ and $t_{f}$ (13 ms), and 11 km/h gave the smallest absolute bias, followed by 9 km/h and 13 km/h (Table 1). The three variables reported a significant negative proportional bias at all speeds and the proportional bias of $t_{f}$ was larger than that of $t_{c}$ (Table 1).

Repeated measures ANOVA depicted significant effects for both methods and running speed, as well as an interaction effect for $F_{z,\max}$, $t_{c}$, and $t_{f}$ (*p* ≤ 0.002; Table 2). Holm post hoc tests yielded significant differences between $F_{z,\max}$, $t_{c}$, and $t_{f}$ obtained using the GSM and IMUM at all speeds (*p* ≤ 0.006). The average RMSE over running speed was 0.15 BW for $F_{z,\max}$ (6%), while it was 20 ms for $t_{c}$ and $t_{f}$, corresponding to 8% and 18%, respectively (Table 2). Cohen’s *d* effect sizes were small for $F_{z,\max}$ and moderate for $t_{c}$ and $t_{f}$, except at 13 km/h which was large for the three variables (Table 2). The average SRD over running speed was 0.28 BW for $F_{z,\max}$ (11%), while it was 32.0 ms for $t_{c}$ and $t_{f}$, corresponding to 13% and 30%, respectively (Table 2).

## 4. Discussion

According to the first hypothesis, an RMSE equal to 0.15 BW was reported for $F_{z,\max}.$ Moreover, according to the second hypothesis, an RMSE equal to 20 ms was obtained for $t_{c}$ and $t_{f}$. Our findings demonstrated systematic and proportional biases, as well as significant differences between gold standard and estimated $F_{z,\max}$, $t_{c}$, and $t_{f}$ at each speed employed. Nonetheless, systematic biases averaged over running speeds were small (0.07 BW and 13 ms) and the RMSEs were smaller than the SRDs, indicating no clinically important difference between the GSM and IMUM. Hence, the present findings support the use of the IMUM to estimate $F_{z,\max}$, $t_{c}$, and $t_{f}$ for level treadmill runs at low running speeds.

A systematic bias of 0.07 BW and an RMSE of 0.15 BW (6%) were reported for $F_{z,\max}$. These errors seemed to be comparable to those obtained using a 10 Hz low-pass cutoff frequency [24], though the bias and RMSE were not explicitly reported [~0.15 BW by visual inspection of the fourth figure in [24] (14–19 km/h)]. In addition, the RMSE found for $F_{z,\max}$ in the present study was equal to the RMSE obtained using two different machine learning algorithms (linear regression and quantile regression forest) [25]. This result suggests that combining IMU data with machine learning algorithms seems to not necessarily be advantageous to estimate $F_{z,\max}$. Using inertial sensors placed on the legs along the tibial axis, Charry, et al. [53] obtained a 6% error on $F_{z,\max}$ (6–21 km/h), while Wouda, et al. [54] achieved a 3% error (10–14 km/h) by using three IMUs (two on lower legs and one on pelvis) and two artificial neural networks. Besides, an RMSE ≤ 0.17 BW was reported when estimating $F_{z,\max}$ using 3D kinematic data of the center of mass or sacral marker trajectory [26]. An RMSE close to 0.15 BW was reported by Pavei, Seminati, Storniolo and Peyré-Tartaruga [27] when the whole-body center of mass acceleration, obtained using kinematic data to estimate $F_{z,\max}$ for running speeds ranging from 7 to 20 km/h, was used for a single participant. Thus, the errors reported for $F_{z,\max}$ in the present study were comparable to those obtained using previously published methods [24,25,26,27,53,54]. Moreover, the RMSE of $F_{z,\max}$ was smaller than its SRD for each tested speed (Table 2), indicating no clinically important difference between $F_{z,\max}$ values obtained using the GSM and IMUM.

The IMUM reported a systematic bias of 13 ms and an RMSE of 20 ms (8%) for $t_{c}$ (Table 1 and Table 2). These errors seemed to be smaller than those obtained using a 5 Hz low-pass cutoff frequency [24], though the bias and RMSE were not explicitly reported [~30 ms by visual inspection of the fifth figure in [24] (14–19 km/h)]. The IMUM employed in the present study might be advantageous compared to that previously used [24] because the present IMUM utilized a single low-pass cutoff frequency (5 Hz) to estimate both $F_{z,\max}$ and $t_{c}$ while the previous method required two different cutoff frequencies (10 Hz for $F_{z,\max}$ and 5 Hz for $t_{c}$). However, the present errors were much higher than those reported by Lee, Mellifont and Burkett [31] (0 ms). These authors used specific spikes in an unfiltered forward acceleration signal recorded by a sacral-mounted IMU sampled at 100 Hz to detect FS and TO events. However, these spikes were not present in most of the data recorded in the present study [see the first figure in in Lee, Mellifont and Burkett [31] vs. Figure 1 herein]. One possible explanation could be that the 10 national level runners recruited by these authors shared a very similar running pattern with specific acceleration spikes that were not always observed in the present study. As a side note, the anterior–posterior acceleration signal recorded by the IMU (Figure 1) was quite different from that depicted in Lee, Mellifont and Burkett [31], and both anterior–posterior IMU signals were different from that assessed using the gold standard anterior–posterior ground reaction force signal [55]. This was also previously observed when reconstructing the anterior–posterior acceleration signal using 3D kinematic trajectories [27]. Besides, the 20 ms RMSE obtained in our study is almost two times larger than the 10 ms RMSE reported by Alcantara, Day, Hahn and Grabowski [25]. Such a difference might be explained by the fact that these authors predicted $t_{c}$ using two different machine learning algorithms (linear regression and quantile regression forest) while the present study estimates $t_{c}$ directly from the post-processing of the vertical acceleration signal recorded by the sacral-mounted IMU. Moreover, such a difference suggests that combining IMU data with a machine learning algorithm may improve the estimations of $t_{c}$ and $t_{f}$ compared to those obtained using IMU data alone. However, the robustness of the machine learning algorithms employed by these authors might be questioned as these algorithms were trained on 28 runners and tested on 9 runners, which is below the median value of 40 participants used for this kind of research question [56]. Nonetheless, further studies would be required to evaluate if applying a machine learning algorithm on our IMU data, which contains 100 participants, would be more accurate in estimating $t_{c}$ and $t_{f}$. Using foot-worn inertial sensors, the systematic bias on $t_{c}$ was ~10 ms (10–20 km/h) [33] and the RMSE was ~10 ms (11 km/h) [38]. Hence, the errors reported for the IMUM [systematic bias: 7 ms; RMSE: 15 ms (6%) at 11 km/h] were comparable to those previously reported using foot-worn inertial sensors. In addition, Falbriard, Meyer, Mariani, Millet and Aminian [33] reported a proportional bias for $t_{c}$, as in the present study. Using 3D kinematic data, the RMSE was larger or equal to 15 ms for $t_{c}$ (20 km/h) [46] while using a photoelectric system, a bias of ~1 ms was reported for $t_{c}$, though validated against motion capture (12 km/h) [57]. Therefore, the error reported for the IMUM when estimating $t_{c}$ was comparable to the error obtained using an optoelectronic system [46], but was much larger than the error obtained using a photoelectric system [57]. However, even though these two systems can be used outside the laboratory [20,21], they suffer from a lack of portability and do not allow continuous data collection. For this reason, using a single IMU was advantageous by its portability, and was shown to be quite accurate to estimate $t_{c}$, and therefore $t_{f}$. Indeed, when the error is calculated for many running steps, as $t_{c}$ and $t_{f}$ are based on the same TO events, the bias of $t_{f}$ is the negative of the bias of $t_{c}$, and the RMSEs for $t_{c}$ and $t_{f}$ are mostly the same in absolute (ms) units. Furthermore, $t_{c}$ and $t_{f}$ reported smaller RMSEs than their corresponding SRDs for each tested speed (Table 2), indicating no clinically important differences between $t_{c}$ and $t_{f}$ values obtained using the GSM and IMUM.

A significant effect of running speed was observed for $F_{z,\max}$, $t_{c}$, and $t_{f}$ (Table 2). Moreover, the most accurate estimation was given at 11 km/h (Table 1 and Table 2). These findings could not readily be explained. However, further studies should focus on testing several slower and faster running speeds to further decipher the running speed effect. Then, future studies could focus on constructing a more sophisticated model, considering the running speed to try to improve the estimations of $F_{z,\max}$, $t_{c}$, and $t_{f}$.

A few limitations to this study exist. The IMUM was compared to the GSM only at low running speeds during treadmill runs. However, the IMUM might also perform well overground because spatiotemporal variables between treadmill and overground running are largely comparable [58], although controversial [59]. Nonetheless, further studies should focus on comparing the IMUM to the GSM using additional conditions (i.e., faster speeds, positive and negative slopes, and different types of ground). Moreover, kinetic and IMU data were not exactly synchronized. Therefore, further studies should focus on synchronizing these data and performing FS and TO events comparisons between the GSM and IMUM. This may be useful if the assessment of metrics at specific FS and TO events is needed, e.g., knee angle at FS, using additional IMUs [60] synchronized with the sacral-mounted IMU providing FS and TO events.

## 5. Conclusions

This study estimated $F_{z,\max}$, $t_{c}$, and $t_{f}$ using the vertical acceleration signal recorded by a single sacral-mounted IMU, which was filtered using a truncated Fourier series to 5 Hz. The comparison between the GSM and IMUM depicted an RMSE of 0.15 BW for $F_{z,\max}$, and of 20 ms for $t_{c}$ and $t_{f}$, and small systematic biases of 0.07 BW for $F_{z,\max}$, and 13 ms for $t_{c}$ and $t_{f}$ (average over running speeds). These errors were comparable to those obtained using previously published methods. Moreover, the RMSEs were smaller than the SRDs, indicating no clinically important difference between the GSM and IMUM. Therefore, the findings of this study support the use of the IMUM to estimate $F_{z,\max}$, $t_{c}$, and $t_{f}$ for level treadmill runs at low running speeds, especially because an IMU has the advantage to be low-cost and portable, and therefore seems very practical for coaches and healthcare professionals.

## Figures and Tables

**Figure 1 sensors-22-00784-f001:**
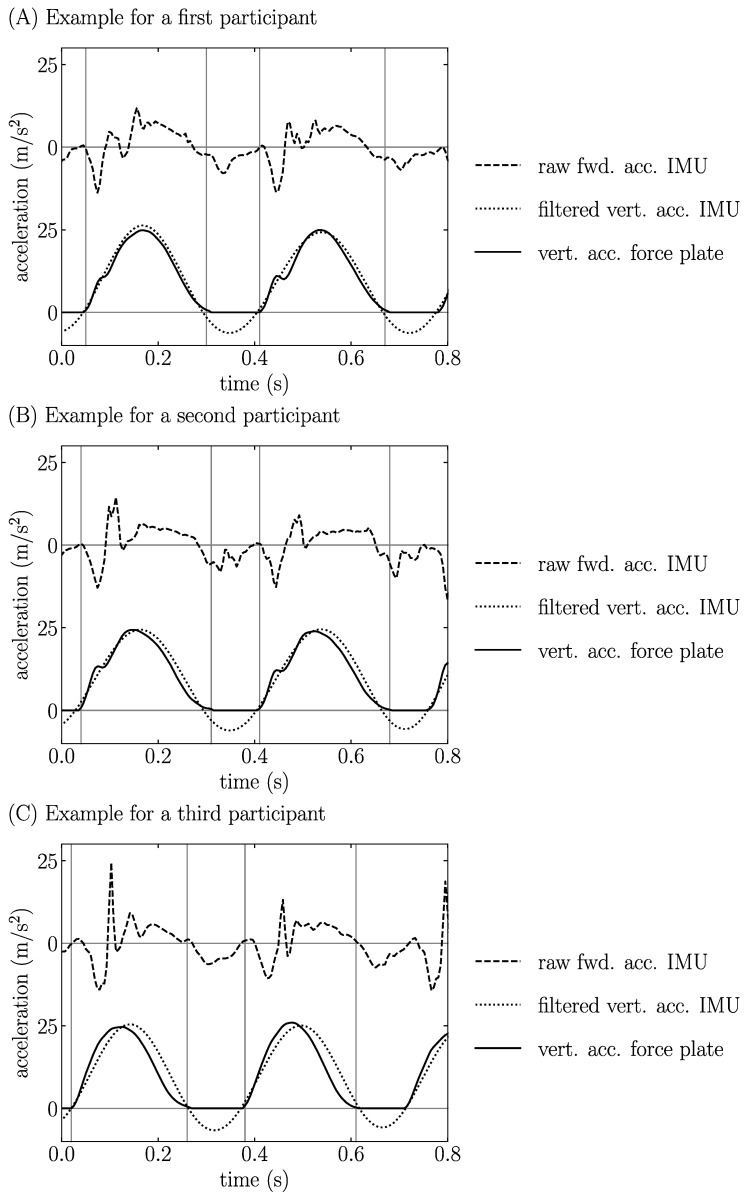
Raw forward acceleration from inertial measurement unit (IMU), filtered vertical acceleration from IMU, and vertical acceleration from force plate during a running stride for three representative participants at 11 km/h are reported in (**A**–**C**). The vertical lines represent foot-strike and toe-off events as determined using a 20 N threshold on force plate data.

**Table 1 sensors-22-00784-t001:** Systematic bias, lower limit of agreement (Lloa), upper limit of agreement (Uloa), and proportional bias ± residual random error together with its corresponding *p*-value between peak vertical ground reaction force ($F_{z,\max}$), contact time ($t_{c}$), and flight time ($t_{f}$) obtained using inertial measurement unit method and gold standard method at three running speeds. Confidence intervals of 95% are given in square brackets [lower, upper]. Significant (*p* ≤ 0.05) proportional biases are reported in bold font.

	Speed (km/h)	Systematic Bias	Lloa	Uloa	Proportional Bias (*p*)
$F_{z,\max}$ (BW)	9	0.05 [0.04, 0.05]	−0.21 [−0.22, −0.20]	0.30 [0.29, 0.31]	**−0.28 ± 0.02 (<0.001)**
	11	−0.04 [−0.04, −0.03]	−0.31 [−0.32, −0.30]	0.23 [0.22, 0.24]	**−0.41 ± 0.02 (<0.001)**
	13	−0.13 [−0.13, −0.12]	−0.45 [−0.46, −0.43]	0.19 [0.18, 0.20]	**−0.51 ± 0.02 (<0.001)**
$t_{c}$ (ms)	9	−9.9 [−10.6, −9.1]	−43.7 [−45.0, −42.4]	23.9 [22.6, 25.2]	**−0.38 ± 0.02 (<0.001)**
	11	7.3 [6.5, 8.0]	−24.6 [−25.8, −23.4]	39.1 [37.9, 40.3]	**−0.37 ± 0.02 (<0.001)**
	13	20.2 [19.5, 20.9]	−10.1 [−11.3, −9.0]	50.6 [49.4, 51.7]	**−0.29 ± 0.02 (<0.001)**
$t_{f}$ (ms)	9	9.9 [9.1, 10.7]	−23.8 [−25.0, −22.5]	43.5 [42.3, 44.8]	**−0.79 ± 0.02 (<0.001)**
	11	−7.4 [−8.1, −6.6]	−39.2 [−40.5, −38.0]	24.5 [23.3, 25.8]	**−0.86 ± 0.02 (<0.001)**
	13	−20.4 [−21.1, −19.7]	−50.8 [−52.0, −49.7]	10.0 [8.9, 11.2]	**−0.91 ± 0.02 (<0.001)**

Note: For systematic biases, positive and negative values indicate the inertial measurement unit method overestimated and underestimated $F_{z,\max}$, $t_{c}$, and $t_{f}$, respectively.

**Table 2 sensors-22-00784-t002:** Peak vertical ground reaction force ($F_{z,\max}$), contact time $\left(t_{c}\right)$, and flight time ($\left(t_{f}\right)$) obtained using the gold standard method (GSM) and inertial measurement unit method (IMUM) together with the root mean square error [RMSE; both in absolute (ms or BW) and relative (%) units], as well as Cohen’s *d* effect size and smallest real difference (SRD) for three running speeds. Significant (*p* ≤ 0.05) method of calculation, running speed, and interaction effect, as determined by repeated measures ANOVA, are reported in bold font. * Significant difference between $F_{z,\max}$, $t_{c}$, and $t_{f}$ obtained using the GSM and IMUM at a given running speed, as determined by Holm post hoc tests.

Speed (km/h)	Parameter	$$F_{z,max}\;(\mathrm{BW})$$	$$t_{c}\;(\mathrm{ms})$$	$$t_{f}\;(\mathrm{ms})$$
	GSM	2.37 ± 0.19 *	278.3 ± 22.2 *	92.8 ± 22.4 *
9	IMUM	2.42 ± 0.14	268.4 ± 15.5	102.7 ± 10.8
	RMSE (absolute)	0.13	18.5	18.6
	RMSE (%)	5.3	6.7	20.1
	*d*	−0.27	0.49	−0.54
	SRD	0.26 (11%)	33.8 (12%)	33.7 (36%)
	GSM	2.51 ± 0.19 *	249.7 ± 19.2 *	111.5 ± 19.7 *
11	IMUM	2.47 ± 0.13	256.9 ± 13.9	104.1 ± 9.1
	RMSE (absolute)	0.13	16.4	16.5
	RMSE (%)	5.1	6.6	14.8
	*d*	0.22	−0.41	0.45
	SRD	0.27 (11%)	31.8 (13%)	31.9 (29%)
	GSM	2.62 ± 0.20 *	227.6 ± 16.5 *	122.8 ± 17.5 *
13	IMUM	2.49 ± 0.11	247.8 ± 12.8	102.4 ± 8.0
	RMSE (absolute)	0.19	24.4	24.5
	RMSE (%)	7.4	10.7	20.0
	*d*	0.73	−1.26	1.37
	SRD	0.32 (12%)	30.3 (13%)	30.4 (25%)
**Method of calculation effect** **Running speed effect** **Interaction effect**		***p* = 0.002**	***p* < 0.001**	***p* < 0.001**
		***p* < 0.001**	***p* < 0.001**	***p* < 0.001**
		***p* < 0.001**	***p* < 0.001**	***p* < 0.001**

Note: Values are presented as mean ± standard deviation.

## Data Availability

The datasets supporting this article are available on request to the corresponding author.

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
