# Peer review of "A Single Sacral-Mounted Inertial Measurement Unit to Estimate Peak Vertical Ground Reaction Force, Contact Time, and Flight Time in Running"

_sensors, 2022, doi:10.3390/s22030784_

Round 1

Reviewer 1 Report

This manuscript did not write well. I do not recommend publishing this manuscript in Sensors. There is only one figure to describe in the manuscript. It is hard for me to get the main idea of this topic. I even do not know how the authors get the data in the introduction section. I do not think this is a good manuscript to publish.

Author Response

Please, find the response to the comments in the word document attached.

Reviewer 2 Report

General Comments:

The authors present an approach that uses a single sacral-mounted IMU, a truncated Fourier series and simple force threshold to make estimations of contact and flight time, and peak vertical force. This manuscript appears to be an incremental extension of the authors’ previous work, presenting findings that show moderate improvement in RMSE for estimation of these measures, as compared to their own recent publication. In the Introduction, the purpose statement says that a simple change to the detection threshold would be used to allow contact and flight time detection. The hypothesis statement then goes on to say that the modified detection threshold will result in smaller RMSE than the authors’ previous work for contact and flight time and a smaller RMSE for peak vertical force as compared to Alcantara et al (Ref #25).

The work presents findings from the same apparent sample (n=100) as that published this year (Ref #24) in the Journal of Biomechanics (vol. 127). It is possible that it is a slightly different sample, since Ref #24 lists 26 females out of the 100, and the current manuscript lists 27 females out of the 100. The similarity with the Journal of Biomechanics article somewhat calls into question the originality of the currently presented work.

In addition to the above concerns, the authors seem to refer heavily to Day et al (Ref #22) and Alcantara et al (Ref #25), for comparison of RMSE of the same outcome variables. Considering the recent publication of the previous Patoz et al work (Ref #24), and the close comparisons to the Day et al and Alcantara et al publications, it would seem that the authors have only slightly modified their approach with a simple detection threshold adjustment to make an incremental improvement to their work. It is questionable whether the adjustment now warrants an entirely new manuscript.

The reviewer does note that the authors have expanded their previous analysis to include estimation of peak vertical force. This is a larger improvement to their previous work. However, it is unclear whether the method presents anything innovative or inherently different than what was presented in Alcantara et al. The hypothesis statement in the current manuscript states that they use a simple method which does not rely on machine learning, implying that Alcantara solely utilized a machine learning approach. Upon a closer read of the Alcantara paper, the authors will note that two general approaches were compared; machine learning and linear regression.

It seems that the technique for estimating peak vertical force presented in the current manuscript is essentially the same as that reported in a Gait & Posture paper by the same group (Patoz et al., 2021, vol. 89). In that publication, the authors report from a larger sample (n=115), and report findings using a single marker and the COM rather than an IMU, but otherwise, it seems to use the same force estimation approach.

The concerns stated above should not be weighted too heavily. It is common for research groups to build upon their own work. However, it is imperative that the authors more clearly clarify what is different between the current work and that of the previously published work. Otherwise, the large number of publications coming from this single data set can serve to confuse the reader as to whether this is the same analysis repeated multiple times with incremental changes, or whether it builds on the previous work. There is a risk of the appearance of re-published work.

The written presentation is good. However, there are numerous instances within the manuscript where the syntax and use of the English language is poor. The authors should seek out an English language editor to assist with syntax and appropriate word selection.

Specific Comments:

Abstract –

  • Line 17; ‘portative’ should be ‘portable’
  • Line 19; The purpose of the work “was to modify an already existing method”. This statement raises the question about whether the approach described in the paper is really novel, or just an iterative improvement. Regardless, in the abstract the authors need to more explicitly state what was modified, in comparison to the previous approach, and what specifically was improved, or what was the same as before. This is covered a bit more precisely in the main body of the manuscript, although not quite to an acceptable level yet.
  • Sentence syntax and wording choices are rough throughout the abstract. Please consult with a native English speaker for phrasing.

Introduction –

  • Line 35; Poor English usage here. The phrase ‘allowed to distinguish’ does not make sense in this sentence.
  • Line 45; Should be ‘portable’, not ‘portative’
  • Line 49; There should either be a ‘the’ before ‘IMU’ or the IMU should be plural (IMUs).
  • Lines 51-53; Poor English here. ‘permitted to identify’ is not correct phrasing.

Methods –

  • Lines 143-144; Why was interpolation needed for the 3D force data? Was it not collected in a continuous fashion, using an A/D conversion? This approach needs clarified and justified, as it seems non-standard, at least as currently worded.
  • Lines 147-151; The phrasing in this paragraph is poor. Suggest using ‘defined by’ in place of ‘given by’.

Results –

  • Line 190; Poor phrasing here. It is not clear what is meant by ‘the one of’.

Discussion –

  • Line 252; Poor phrasing again here. ‘the 11ms one’ is not correct phrasing.
  • Lines 289-290; Poor English phrasing here. ‘similar error than previous’
  • Lines 303-304; From this statement, it appears that the authors did not synchronize the signals. This is concerning. Perhaps I missed it in the Methods section, but I don’t recall this being mentioned. At minimum, it should be mentioned in the Methods. If there were truly no synchronization efforts made, then it should be clarified how the authors are making temporal comparisons.

Conclusion –

  • Lines 310-315; Poor English usage in this paragraph. Specifically, in line 310; ‘used a IMUM based one’, and again in line 314; ‘are similar than’

Author Response

(The authors gave the same response as above.)

Reviewer 3 Report

Main comments

The authors aimed to analyze the accuracy of an already existing method that estimates effective contact and flight times from data recorded using a single sacral-mounted IMU. 100 Runners ran at three constant speeds (9, 11, and 13km/h), and IMU data (208Hz) and force data (200Hz) were acquired by a sacral-mounted IMU and an instrumented treadmill. The measure errors were determined, and a qualitative comparison was done. The authors found out that the proposal has similar error compared with gold-standard determination (force platforms).

The study is very interesting and surely has application potential with high ecological validity. Coaches and health professionals working with running interventions could benefit from this study.

My main concerns on the study are:

- The abstract is vague on the main objective of the study. Which alteration was done?

- Consider using the abbreviation already extensively used to make the reading clearer, for example Tce and Tae (please see the Cavagna’s work) instead of Tc and Tf, to clarify that you are analyzing the effective times.

- lines 62-64 - I suggest to improve the explanation on the importance of measure the effective instead of traditional/real times in running, to understand the principles underlying the elastic mechanism of running (for example, showing that effective times were already observed as more sensitive to differences in terms of running performance than Tc and Tf).

- the authors ignore important production on previous attempts using kinematic data to estimate GRF (https://doi.org/10.1016/j.jsams.2018.12.007 and https://doi.org/10.1123/jab.2015-0329) the last study estimated the effective times.

- endurance speeds is somewhat forced, I suggest denominates as low running speeds.

- Consider using modern techniques of instrument/procedures validation as follows:

I suggest, in summary calculate:

A receiver-operator characteristic curve to determine the cuff off point which optimally defined the Minimal practical/clinical important difference based on sensitivity and specificity values for each observed change value. You might calculate the MCDI from your own data as function of speed and see the trade-off between sensitivity and specificity of technique.

a stratifeying  the analysis according mean own data influencing the likelihood of achieving the MCID and odds ratios.

Consider following the STRAD reporting guideline for validation studies:

https://www.equator-network.org/reporting-guidelines/stard/

- lines 237-242 - please, respond clearly your hypotheses here.

- observing the figure 1, clearly we see that the IMU is far to reach the crucial values as vertical impact peak and rate of force development (which you used as justification for the study). Perhaps, comparing the RFD in fore-foot striker could take valid information.

- please, the reference is overused in all paper. For example, consider using more original references to some statements as in lines 100-102. Certainly, they did not originality in this statement (even being interesting).

-lines 124-125 - How the sensor was attached?

- even secondary, observing the figure 1, we see that horizontal data is so much far than any valid information (corroborating with 3D kinematics attempts as in https://doi.org/10.1123/jab.2015-0329)

Author Response

(The authors gave the same response as above.)

Reviewer 4 Report

I recommend the publication of the study in present form

Author Response

(The authors gave the same response as above.)

Round 2

Reviewer 1 Report

It seems that the authors missed the experimental section. Also, the authors need to make more figures to complete this manuscript. The current one is not completed in the current form. Please make it more clear for readers and more data to show your progress.

Author Response

-
